# Use of Twitter during Televised Election Debates: Spanish General Election (28 April 2019) vs. French General Election (24 April 2022)

Julia Fontenla-Pedreira *, Carmen Maiz-Bar  and Talia Rodríguez-Martelo

Department of Audiovisual Communication and Advertising, School of Communication, University of Vigo, 36005 Pontevedra, Spain
* Correspondence: julia.fontenla.pedreira@uvigo.es

**Abstract:** Social media have become key in political communication, playing a crucial role in election campaigns due to their fast, ubiquitous communication. This paper focuses on the comparison of the use of the social network Twitter in Spanish and French public and commercial television stations, during the last televised debates held during their general elections (2019 and 2022). It seeks to find whether conversation and interaction with their audiences take place, and whether these meet the dialogic principles set forth by Kent and Taylor adapted to Twitter by Ribalko and Seltzer to include usefulness of information, generation of return visits and dialogic loop preservation. To do this, the content of the general Twitter profiles of two French television stations, together with their profiles focused on informative content, were analyzed before, during and after the televised election debate held on 20 April. Likewise, the Twitter profiles corresponding to two Spanish television stations, together with the profiles corresponding to their news programs, were studied before, during and after the televised election debates held on 22 and 23 April. After screening all their posts, those referring to the debate and generating the largest engagement figures were selected, in order to compare the topics covered in the televised debates with those covered in Twitter. The results reveal that the information-focused accounts originate more posts whose content is linked to the televised debates, in contrast with the general accounts. Furthermore, both the unidirectionality of their content, and the lack of dialogue and interaction between these accounts and their audiences, become apparent, in addition to the minimal occurrence of "debate about the debate" flow among users.

**Keywords:** election debates; election campaigns; political communication; social media; Twitter; France2; TF1; FranceInfo; TF1Info; RTVE; Atresmedia; Telediario; Canal24h; Antena3Noticias; laSextaNoticias



## 1. Introduction and Literature Review

### 1.1. Political Communication within the Digital Environment

The French scenario surrounding digital communication media is different from many other countries. First, digital-born media are more prominent than in other European countries. Likewise, legacy media, particularly newspapers, have a historically weaker role in countries such as Germany or the United Kingdom [1]. Printed newspapers have had a more limited reach than in most of Northern Europe and, although television is still the most important and most frequently used news source in that country, the general audience has certainly aged [2]. Furthermore, since the emergence of social media, French citizens have turned to these from the start, and currently 80% of them state that they receive news via the social media, particularly Twitter and Facebook.

Bearing in mind this context and the characteristics of the digital media, such as their versatility, it can be argued that they have instantly turned into a useful medium to develop communications and modern political campaigns. Websites have become obsolete, given that they only allow connections with informative pages, unilaterally fed by the specific

political party, while social media have reached potential voters, who are not interested in politics and do not visit political blogs or websites.

The use of social media has, with greater ease, led internet users to interact with politicians, political clips, or any kind of propaganda, by means of resources provided by the social media: image galleries, stories, reels or any other post format. An internet surfer who is connected for leisure purposes to a video sharing platform, such as YouTube, Instagram or TikTok, for instance, will be eager to see the most popular posts. These videos, provided that they are well designed and labelled, may receive a high score in the social media classification system [3]. This way of reaching voters was one of the specific strengths of the 2008 Barack Obama campaign, when his so-called "friends" uploaded YouTube videos with entertaining content to boost his campaign, such as "Obama Girl" or "I have a crush on Obama", which politically uncommitted American citizens watched in order to share and have a good time [4]. At the same time, different music videos, some of the specifically propagandistic clips such as "Yes we can!", prepared by Will.i.am (a member of Black Eyed Peas), engaged Obama's supporters in the same way that the website NSTV had engaged Nicolas Sarkozy's supporters one year earlier, in France. Nowadays, political online resources available to social media users are more and more elaborate, captivating both politically committed users, and others who were undecided but curious.

The Debate on Twitter across Europe

Elections in different European countries are good illustrations of the way election campaign models are evolving, as well as the role of new technologies in this evolution, and their capacity to create new proposals and information pathways [5]. Several studies account for the use of Twitter in elections to institutions such as the European Parliament, or even its use by members of different parliaments [6,7]. According to Daniel, Obholzer and Hurka [8], parliament members mainly tweet during the weeks when parliamentary and committee meetings are held, and "high district magnitudes and preferential voting lead to less frequent communication" via that specific social network [9]. The same authors add that both larger voter figures in a constituency and the general prevalence of social media in a particular country positively correlate with more active members of parliament on Twitter. Furthermore, they also state that younger politicians—and their parties—are more likely to be active on Twitter, and that, when considering their ideology, "green" parties tweet the most, while far right parties tweet the least.

From a different perspective, authors such as Larsson determine that the degree of activity of politicians on Twitter is closely related to their age, being the youngest the ones who tweet the most. Larsson also indicates that their activity increases along with the number of users that they follow and the percentage of individuals using the internet in their countries [10].

Other studies cover the goals behind the politicians' use of Twitter; for instance, a qualitative study by Frame and Brachotte [11] includes interviews with five French politicians and determines that Twitter was mainly used "to interact with voters, journalists, stakeholders and other politicians and to disseminate information" (p. 1), as well as to monitor current events or the public opinion. As for Spain, studies carried out by López-Meri, Marcos-García and Casero-Ripollés [12] claim that politicians use Twitter mainly to promote their policy proposals.

Research papers from Germany, such as the one by Stier et al. [13] on Twitter and Facebook, point out that politicians use Facebook to provide detailed communications, to mobilize voters, and for longer posts including photos and videos, while Twitter is used for disseminating news, keeping in touch with journalists and quickly reacting to events.

The studies published by Haman and Školník on the use of Twitter by politicians are also of interest, and reveal that most members of national parliaments, in most European countries, own profiles on this social media platform: almost every member of the West European parliaments has an active Twitter account. On the contrary, parliament members in post-communist countries seem to be less interested in Twitter and tend to use other

communication platforms; "exceptions can be found both in the West (Portugal and small monarchies) and in the East (Poland)" (p. 8).

Haman and Školník also state that the politicians' activity level is higher in Western European countries and lower in post-communist countries, being those levels closely related to the percentage of Twitter users in each country.

### 1.2. Election Debates in France: Before and After

Election debates have been studied from different perspectives throughout history, focusing on aspects such as their regulation [14], development, formats and staging [15], media interest [16], or information management and verification [17]; and have varied in their characteristics depending on the parliamentary systems of each country. In Western Europe, for instance, election debates were introduced in television programming almost at the same time as in the USA, becoming true media events. The one-on-one duel confrontation format has generated expectations for an exciting event, in which entertainment values interfere, and even rule, over the political and informative functions of the debates [18].

In France, ever since the first debate between François Mitterrand and Valery Giscard d'Estaing, held in 1974, televised debates between the two candidates to the final round of the elections have become one of the main events in the presidential campaigns. Likewise, the strong correlation between the best performance in the debate, and the results of the election, made these debates seem decisive. But looking back into past French presidential debates, it becomes apparent that, most times, they did not modify the election results, as compared to previous survey results. It seems that the leader in those surveys usually performs better in the debate, as compared to his/her rival. 2017 was particularly relevant in this aspect, given that not one, but 11 televised debates took place, including, for the first time, two debates before the first round of the elections [19]. The same author highlights that the role of television during the campaigns, particularly televised debates, is far from vanishing, even with the rise of Internet and social media.

The use of new digital trends underwent a major turnaround in the way election debates were understood in France. The negative result of the 2005 European Constitution election was partly attributed to the digital campaign, which received more attacks than compliments; these attacks were led by blogs, created by anonymous citizens, while most traditional and political media rooted for a positive vote [20].

It was in 2007 when presidential candidates Ségolène Royal and Nicolas Sarkozy created complete websites, and the 2012 presidential campaign confirmed that the political communication specialists used all the facilities of the social media, that is, they sought to take advantage of interactivity in all its possible forms [21]. The victory of Ségolène Royal in the socialist primary elections was a consequence of the penetration of internet into politics: a system of "friendly" blogs was implemented, attached to her main website, "Désir d'avenir". On the side of Nicolas Sarkozy, her main rival, a Web-TV, NSTV.com (Nicolas Sarkozy Télévision) received high audience shares during the campaign, and became a new internet political television model (Maarek, 2014). This reinforced the idea that internet television was an unavoidable communication tool in most countries with democratic elections, while television is still the main channel for citizens to receive their political information [22].

### 1.3. Election Debates in Spain: Before and After

The current Spanish environment is characterized by the breakaway from a two-party system and by political digital communication strategies [23–26]. These turn the debate into an essential tool, both for political parties and for society and democracy.

In Spain, from 1993 and up to now, a total of 12 televised election debates have been held among presidential candidates; nevertheless, if we bear in mind that the party leader Mariano Rajoy was not involved in two of them, 10 debates should really be considered [27].

This is a very low number, taking into account that, from the first democratic elections to the first televised election debate, in 1993, 16 years had gone by. 15 more years would

have to pass until the next debate, between the PP candidate, Mariano Rajoy, and the PSOE candidate, Rodríguez Zapatero, was held in 2008 [28].

The new party system led to a multi-party environment in 2015 [29], modifying the format of the debates, opening them to more candidates. This went from the traditional "one-on-one" between PP and PSOE, to a debate with four participants when Podemos and Ciudadanos were included, and then to five participants when Vox burst onto the scene—which gave rise to disturbances, obstacles and challenges for traditional parties and for the Junta Electoral Central (JEC—Central Election Board), the organization in charge of settling the disputes caused by this new political environment.

Castromil and Rodríguez [30] state that, when considering the two traditional parties (PP and PSOE), it was the PP that rejected the new debate format, as Mariano Rajoy declined the invitation to participate in a debate with Sánchez, Iglesias and Rivera, requesting to go back to the traditional "one-on-one" and leaving the emergent parties aside. This changed when the PP realized that the four-participant format could ran the risk of becoming a three-participant format, so they went for an intermediate format. On the one hand, they wanted to avoid the costs of not being present in the debate, and on the other hand, they did not want to discredit Rajoy, who was opposed to the new format. The solution to the problem was to send the government vice-president, Soraya Sáez de Santamaría, to the debate.

Similarly, when the 2019 campaign took place, the current president Pedro Sánchez decided to attend the debate held by the privately-owned station Atresmedia, including the new party Vox (a five-participant format that was finally forbidden by the JEC), instead of accepting the proposal from the public station RTVE, which did not involve Vox, as his strategy involved demonstrating the possible pact and coincidence of interests between Vox, PP and Ciudadanos [31]. Finally, two debates were held back-to-back: April 22nd on RTVE and April 23rd on Atresmedia.

*1.4. Televised Election Debates and Interaction in Social Media*

Traditional audiences have been transformed, resulting in the need to examine new formats for the relationship between television and its audiences [32]. For that reason, new metrics have been incorporated into the analysis of the television environment, as different measurement data must be taken into account [33]. These new metrics pose a challenge for the communication strategies of both political parties and television stations [34], which leads to new ways of generating content and engaging users.

One of these social media metrics is, specifically, engagement, initially considered as the attention towards an object and the symbolic representation of subjective mental states [35], via social media tools (mainly "like", "share" and "comment"). These may be recorded and classified for their subsequent measurement. Their use gave rise an academic debate regarding their political adaptation as a user response index based on comments in social media, as well as a quantitative tool for measuring audience participation and interaction with digital content.

Different authors suggest combined formulas to calculate engagement indexes, ranging from adding up the number of "likes", "shares" and "comments", divided by the number of messages [36]; dividing the same numerator by the number of followers [37] or using as a denominator the number of messages divided by the number of followers [38]. In any case, social media interactions are not only the occasional ones shared on user profiles, but also the number of times that this content is commented, receives a "like", or is shared by other people. In these interactions, elements such as mentions or hashtags help to increase the dialogue.

Hashtags, for instance, organize debates concerning specific topics or events, structure the digital conversation, and encourage interaction [39], but they are also used and shared by political parties to promote digital mobilization with a real influence on the political agenda, while mentions enable ways to interact with other users. Political hashtags gained momentum in events such as the 2009 Iran presidential election; in fact, #iranelection was

the number one trending topic that year [40]. The value of a political hashtag has its origins in the real-time nature of the shared information.

Furthermore, it is important to highlight that, in the social media era, the strict equality regulations regarding candidate speaking times on television have become obsolete, and the content has become the cornerstone to achieve engagement. This has encouraged political parties from many countries, such as France, to request modifications to the election regulations.

### 1.5. Political Engagement, Intentional or Accidental Exposure?

The possibilities that social networks offer for sharing information and interacting with other users, turn them into individual and personalized spaces, through which any prosumer user can cause different effects on other people, independently of the fact that they belong to their group or friends or are complete strangers. These effects subsequently originate new repercussions, depending on the use that the recipients make of that information [41]. A new range of possibilities for reflection opens up here, from the simple activity consisting on acting as observers or readers, throughout sharing a "like", being the latter the behavior preferred by the audiences regarding political parties, as it involves less pressure and commitment. This is the seed for the debate about whether social media contribute to spreading the emotions regarding political content. The "social buttons" reflect, according to some authors, volatile and weak affective states, and a low-responsibility commitment, different from commenting and sharing information [42], which only has the so-called influencers as its real competitors [43].

Nowadays, political debate is increasingly common in social media. Nevertheless, some moral ideas and feelings spread faster than others, and even in a more detailed way. Expressing emotions is key for disseminating ideas [44], both moral and political, in social media, a process that Knoll et al. call "moral contagion".

Internet users perform different activities depending on their motivations, such as intentional searches for political information. This intentional exposure happens when a person feels the need to receive political information and actively searches for it, deploying all tools within reach. For instance, users may rely on the search function within the social media sites, scan their own information sources looking for that political content [45], look within Facebook news platforms, or visit the profile of a political party or candidate [46].

But they can also "accidentally" find the information, when the content is shared by people within their close circle. This way, a person who is not interested in politics only needs to be connected to a group of politically engaged people to be regularly presented with political data and information.

Furthermore, Knoll et al. state that, in the context of an election campaign, people also want to interact with entertainment content in social media, that is, the seek social interaction without political nuances, feeling the need to find information or expression not related to politics, or work on non-political parts of their identities. Nevertheless, the drive to find non-political expression and to work on non-political aspects of their identity lead to accidental exposures. Obviously both exposure modes, intentional and accidental, are conditioned by different factors: on the one hand, the features of the networks surrounding each user, on the other, the interconnections between their acquaintances, the degree of literacy of the network users, and the social media algorithms.

Bearing in mind the first factor, it is important to mention that users can nowadays actively manage their networks, hiding posts or blocking users, unfriending other people or simply not following them anymore. In this sense, Zhu et al. [47,48] point out that those users with strong political ties filter their news sources in a more active way and, consequently, are exposed to information sources that are more in line with their way of thinking. On the contrary, users with a more heterogeneous network in terms of contacts will also have more varied information, thus increasing the possibilities of participation and political exposure [49].

Considering that intentional exposure involves actively looking for information, Knoll et al. write that this process is influenced by the literacy degree of the internet users regarding digital culture, that is, their ability to access, understand, assess and create content in social networks. Lee et al. [50] argue, for instance, that some users may be more proficient at identifying high quality political information in their search for news, while others, with specific political interests, only come across low-quality information, such as "populist politicians" [51].

In addition, it is important to bear in mind another factor that determines the individual's exposure to political information: the personalization of content. The social media algorithm exposes users to information related to their preferences, or taking into account other shared information, that is, considering his or her previous browsing history, and the information included in his or her profile [52,53]. All this creates a cluster that causes internet users to be more or less exposed to specific political content [54], and with this comes the "affection" towards certain content resulting from this need for entertainment and emotional experiences.

In this sense, Knoll et al. indicate that, depending on the needs and emotional states of individuals, they may intentionally or accidentally expose themselves to political information; for instance, users who experience feelings of anxiety have a tendency to engage with different political views, and intentionally expose themselves to that information [55]. Other authors, such as Lagares et al. [56], point to the importance of emotional framing, particularly negative framing, as a generator of political commitment and political opinion [57,58] and, last but not least, as a political polarizing element [59].

## 2. Objectives and Methods

The general objective (GO1) of this paper is to observe and track, both with quantitative and qualitative perspectives, the presence, audience and conversation in the general Twitter profiles of the publicly owned Spanish and French television stations RTVE and France2, and the privately owned Atresmedia and TF1. Likewise, the profiles corresponding to their news programs (Telediario, Canal24h, Antena3 Informativos, LaSexta Noticias, France Info and TF1 Info) before, during and after the televised election debates held on 22 April—RTVE—and 23 April—Atresmedia (for the 28 April 2019 election), and 20 April 2022 (TF1 and France2) will also be analyzed.

The dates chosen are therefore the day when the debates were held, the day before and the day after, that is, 21–23 April for RTVE; 22–24 April 2019 for Atresmedia; and 19–21 April for the French debate. As for France, the stations were chosen because the election debate was a joint production by both stations.

After screening all their posts, those specifically referring to the debate and generating the largest engagement figures were selected, in order to compare the topics covered in the televised debates with those covered in Twitter. The results reveal that the information-focused accounts originate more posts whose content is linked to the televised debates, in contrast to the general accounts. Furthermore, both the unidirectionality of their content, and the lack of dialogue and interaction between these accounts and their audiences, become apparent, in addition to the scarce occurrence of "debate about the debate" flow among their users.

4539 posts were screened for content using the social listening platform "Twlets". The hypothesis propounded by Fontenla-Pedreira et al., who argued that the public stations do not participate—via social media—in the dialogue and interaction with internet users, and that they only use those platforms for dissemination purposes, and to enable the broadcasting of the content of their programs, not to encourage debate. Taking those previous research papers into account, the following specific objectives (SO) are established:

- Specific Objective 1—SO1: To analyze, from a quantitative perspective, the presence, activity and audience of the content during the day of the debate, the day after, and the day before, on the social media platform Twitter, for the abovementioned accounts, focusing on similarities and differences in their activity.

- Specific Objective 2—SO2: To study whether the posts from the different channels included in the sample generate Twitter conversations, according to some of the dialogic principles set forth by Kent [60] and adapted to Twitter by Ribalko and Seltzer [61], to include usefulness of information, generation of return visits and dialogic loop preservation. Maintaining the dialogic loop enables the determination of conversation and interaction by two means: either asking the users questions, or responding to comments provided by the users.
- Specific Objective 3—SO3: To analyze the inception of agenda-setting, with the purpose of verifying whether the topics and content posted on Twitter by the television stations match the main interests of the users, revealed by their comments in that social network. For content analysis, the five posts with the largest engagement figures were used, selected from each station on the pre-determined dates.

## 3. Results

### 3.1. Quantitative Analysis

Tables 1–3 depict the posts included in the analyzed Twitter profiles, as well as those with debate-related content and their engagement figures. Tables 4–6 detail the candidate mentions, while Tables 7–9 show the use of hashtags and Tables 10–12 illustrate the recorded sentiment-related information.

**Table 1.** Posts including content related to the 20 April 2022 Presidential Debate.

| TV Station | Total Sampled Tweets | Tweets Including Debate-Related Content | Post-Debate Average Engagement |
|---|---|---|---|
| France2 | 152 | 23 | 0.2 |
| TF1 | 380 | 97 | 0.5 |
| France2 Info | 489 | 158 | 0.7 |
| TF1 Info | 563 | 308 | 0.8 |

Source: Prepared by the authors.

**Table 2.** Posts including content related to the 22 April 2019 Presidential Debate.

| TV Station | Total Sampled Tweets | Tweets Including Debate-Related Content | Post-Debate Average Engagement |
|---|---|---|---|
| RTVE | 132 | 35 | 0.3 |
| Telediario | 356 | 110 | 0.6 |
| Canal 24 h | 204 | 112 | 0.5 |

Source: Prepared by the authors.

**Table 3.** Posts including content related to the 22 April 2019 Presidential Debate.

| TV Station | Total Sampled Tweets | Tweets Including Debate-Related Content | Post-Debate Average Engagement |
|---|---|---|---|
| Atresmedia | 289 | 42 | 0.5 |
| Antena3 Noticias | 465 | 143 | 0.7 |
| laSexta Noticias | 663 | 190 | 0.8 |

Source: Prepared by the authors.

**Table 4.** Candidates mentioned—French Debates 22 April 2022.

| TV Station | Macron Mentions | Le Pen Mentions | No Mentions |
|---|---|---|---|
| France2 | 42% | 40% | 18% |
| TF1 | 39% | 41% | 20% |
| France2 Info | 44% | 45% | 11% |
| TF1 Info | 47% | 46% | 7% |

Source: Prepared by the authors.

**Table 5.** Candidates mentioned—Spanish Debates 22 April 2019.

| TV Station | Pedro Sánchez | Pablo Casado | Pablo Iglesias | Albert Rivera | No Mentions |
|---|---|---|---|---|---|
| RTVE | 12% | 17% | 21% | 32% | 18% |
| Telediario | 11% | 16% | 20% | 34% | 19% |
| Canal 24 h | 15% | 19% | 22% | 29% | 15% |

Source: Prepared by the authors.

**Table 6.** Candidates mentioned—Debate 23 April Atresmedia.

| TV Station | Pedro Sánchez | Pablo Casado | Pablo Iglesias | Albert Rivera | No Mentions |
|---|---|---|---|---|---|
| Atresmedia | 10% | 10% | 13% | 37% | 30% |
| Antena3 Noticias | 8% | 17% | 22% | 36% | 17% |
| laSexta Noticias | 12% | 19% | 12% | 32% | 25% |

Source: Prepared by the authors.

**Table 7.** Hashtags used by each TV station—French debate 22 April 2022.

| TV Station | #DebatMacronLePen | #LeDebat2022 | #Présidentielles2022 |
|---|---|---|---|
| France2 | 90% | 92% | 40% |
| TF1 | 91% | 95% | 53% |
| France2 Info | 89% | 90% | 45% |
| TF1 Info | 93% | 96% | 45% |

Source: Prepared by the authors.

**Table 8.** Hashtags used by each TV station—Spanish debate 22 April 2019.

| TV Station | #ELDEBATEenRTVE | #Debate22A | #EleccionesGenerales |
|---|---|---|---|
| RTVE | 93% | 90% | 30% |
| Telediario | 94% | 92% | 33% |
| Canal 24 h | 94% | 89% | 29% |

Source: Prepared by the authors.

**Table 9.** Hashtags used by each TV station—Spanish debate 23 April 2019 Atresmedia.

| TV Station | #ElDebateDecisivo | #DebateAtresmedia | #EleccionesGenerales |
|---|---|---|---|
| Atresmedia | 92% | 91% | 30% |
| Antena3 Noticias | 93% | 94% | 41% |
| laSexta Noticias | 96% | 95% | 32% |

Source: Prepared by the authors.

**Table 10.** Sentiment towards the candidates, measured using user comments—French Debate 22 April 2022.

| TV Station | Sentiment towards Macron | Sentiment towards Le Pen |
|---|---|---|
| France2 | Neutral | Neutral |
| TF1 | Neutral | Neutral |
| France2 Info | Positive | Negative |
| TF1 Info | Neutral | Positive |

Source: Prepared by the authors.

**Table 11.** Sentiment towards the candidates, measured using user comments—Spanish Debate RTVE 22 April 2019.

| | Sentiment | | | |
|---|---|---|---|---|
| TV Station | Pedro Sánchez | Pablo Casado | Pablo Iglesias | Albert Rivera |
| RTVE | Neutral | Neutral | Negative | Negative |
| Telediario | Neutral | Neutral | Negative | Negative |
| Canal 24 h | Neutral | Neutral | Neutral | Negative |

Source: Prepared by the authors.

**Table 12.** Sentiment towards the candidates, measured using user comments—Spanish Debate Atresmedia 23 April 2019.

| | Sentiment | | | |
|---|---|---|---|---|
| TV Station | Pedro Sánchez | Pablo Casado | Pablo Iglesias | Albert Rivera |
| Atresmedia | Neutral | Negative | Positive | Negative |
| Antena3 Noticias | Neutral | Negative | Positive | Negative |
| laSexta Noticias | Neutral | Neutral | Positive | Negative |

Source: Prepared by the authors.

*3.2. Content Analysis: Posts with the Greatest Engagement Figures*

Twitter comments on the posts included in the television stations analyzed are limited, particularly on the days before and after the debate. During the debate itself, the number of comments increases, and the criticism of the candidates towards each other fires up Twitter.

3.2.1. Posts with the Greatest Engagement Figures—French Election Debate 2022

In the case of France, the content of the tweets includes the allegations that both Marine Le Pen (Rassemblement Nacional) and Emmanuel Macron (La République en Marche) level against each other throughout the entire election debate.

The false start of the candidate Marine Le Pen monopolized the users' attention during the first minutes of the debate, wondering whether it would be the main event of the evening. Le Pen, whose turn it was to speak first, started during the initial credits, then and jokingly said "I started too early". User comments were immediate: "Le Pen's false start would involve being disqualified in an athletic competition"; "a false start for Marine Le Pen, who thought that she had been given the floor when credits started. That is called "speaking to the void" . . . #Debat2022".

The users who commented on the profiles analyzed considered the beginning of the debate "slow" and "boring", until the topic of the Ukraine war emerged, causing the first clash between the candidates, who engaged in a true fight. This was one of the contents that raised the interest of the users, highlighting Macron's accusation to Marine Le Pen of "depending on Russian power" "and Mister Putin" due to the fact that she had borrowed

money from a Russian bank: "You talk to your banker when you talk about Russia, that's the problem Madame Le Pen" was the sentence used by Macron to attack the extreme right-wing candidate, referring to the loan that she had received from a Russian bank. Le Pen replied stating that it was a false accusation, showing as evidence one of her tweets supporting Ukraine, written back in 2014 and printed in large format—which caused the first "laughs" among the users. Her words were tweeted by all television stations, and retweeted by many users. She responded that she was "an absolutely and totally free woman".

Twitter users also reacted when Macron accused his opponent of wanting to leave the European Union and never seeing "French leaders defending French interests" in the EU. The fight went on as Le Pen said "Don't fall for conspiracy theories", to which the République En Marche candidate replied "Coming from you, I find it attractive".

Memes also swept the attention of the users, who commented, on the profiles of the television stations analyzed, that Marine Le Pen called Macron the "Mozart of finance", referring to his economic mismanagement during the COVID-19 pandemic. Similarly, other memes referred to the moment when Le Pen accused Macron of being a "climate hypocrite" and supporting "the worst of punitive ecology", while he called her a "climate skeptic" because of the reduction in taxes that would be applied to fossil fuels.

During the two and a half hours of the debate, comments on Twitter focused on the attitudes of both candidates: while Macron was associated with adjectives such as "aggressive", Marine Le Pen was linked to a firmness- and resistance-filled aura. Twitter did not forgive Macron either; his body language during the debate, with both elbows on the table, his chin cupped in his hands and frowning, led to many memes as well: "Your psychiatrist for 2 h, but he still gets the 60€ in the end", "Me when there are no chips left in the canteen #debatemacronlepen #presidential2022".

Furthermore, the comments of the social media users did not sideline any of the candidates, the main sentiment being neutral regarding their words. Nevertheless, the posts assigned the Rassemblement national candidate, Marine Le Pen, a more defensive attitude, while Macron was considered to be more relaxed. Comments indicate that Marine Le Pen presented a friendly, smiling countenance that appeared hypocritical and distant when facing her opponent.

In addition, although to a lesser extent, tweets regarding the attacks on the proposals of their opponents regarding incentives to increase salaries and bonuses also prevailed, mutually accusing each other of leading citizens to believe that those increases would be applied automatically.

Another moment that grasped the interest of users on all of the social networks analyzed of was a new accusation by Le Pen to Macron, stating that his work has been based on "betraying the French spirit and the Republic", and "creating a civil war" based on the ban on the veil in public spaces.

There was no shortage of comments from users towards the two journalists moderating the debate, Gilles Bouleau (TF1) and Léa Salamé (France 2), who limited themselves to controlling the time of the speakers and maintaining the order of the interventions, without interrupting or verifying facts. This restrained position was not at all appreciated by users, and comments were heard on the network through mockery: "Gilles Bouleau has the worst hidden fake job in the world", "the poor time masters". "Gilles Bouleau repeats <We are the keepers of time!> . . . calm down tiger!".

Finally, the tweets most commented by users during the day after the debate revolved around who had won the dispute.

### 3.2.2. Posts with the Greatest Engagement Figures—Spanish Election Debates 2019

In the case of RTVE, the debate held on 22 April 2019, the tweets corresponding to its first minutes concentrated on the arrival of the candidates at the filming set, focusing on the presence of women only from the cleaning sector when filming the first scenes, or the "not formal" outfit of Pablo Iglesias. He was also heavily criticized because of his

luxurious house, given that one of his tweets from 2012 was revived, where he questioned the apartment that the former Ministry of the Economy, Luis de Guindos had purchased; the Unidas Podemos leader had asked internet users "would you entrust the economic policies of your country to someone who spends 600,000€ on a luxury penthouse?".

Likewise, the Ciudadanos and PP candidates, Albert Rivera and Pablo Casado, were the targets of many of the contents created by the users. On the one hand, the objects that Rivera used for his presentations were called "Doraemon's magic pocket", and this, together with the posters that he used, made it easier for social media users to create image mockups with the articles that he used, mentioning his then-girlfriend, the singer Malú, restaurant menus or an ID that users called "the very-Spanish ID". Regarding Pablo Casado, most of the comments concerned the scandal regarding his master's degree from the Juan Carlos I University.

Memes about the President of the Government, Pedro Sánchez, were also present, although they were fewer than those devoted to the other candidates. The most commented topic concerned his trips in the Falcon plane, which he usually uses for his travels.

When focusing on Atresmedia and its debate held on 23 April 2019, the first minutes were also commented, particularly comparing the arrival of the Ciudadanos leader, Albert Rivera, in a high-end Lexus car, while Pablo Iglesias arrived by taxi.

As in the previous debate, Albert Rivera was the target of many comments, due to all the objects that he used in his speeches: "Rivera, an unboxing please"; "Rivera has Diogenes syndrome"; "Is that a lectern or a street market? ". Tweeters also criticized his several interruptions to other candidates' speeches, and the performance of the moderators.

Although Santiago Abascal did not participate in the debate (he was invited by Atresmedia, but the Central Election Board forbade his participation due to proportionality regulations, which depend on the results from previous elections), some of the posts with the largest engagement figures mention the Vox leader, focusing on his statements regarding this absence. His proposal about reinstating a compulsory military service was also highlighted, but comments mostly focused on Abascal's words, such as: "Get the f***k back to work, stop talking about military service and saying you will do things that you haven't done up to now, like working for instance"; "I think the military service idea is ridiculous. But a few months spent working in customer-facing positions would be good for everyone"; "Wow . . . do as I say, not as I do . . . good example".

## 4. Discussion of Results and Conclusions

Taking into account the results of this research paper, and bearing in mind its first specific objective, it becomes apparent that the Twitter profiles with specifically informative content (that is, Telediario, Canal 24 h, Antena3 Noticias and la Sexta Noticias, for Spain; and TF1 Info and France Info, for France) were those which generated the most content regarding presidential debates. When considering the generalist profiles, the number of posts was lower and mostly limited to the day when the debate was held. On the contrary, the news-related accounts did generate a great amount of content, leading to internet user reactions, although the social media managers did not respond to those comments, even if these were disrespectful and should have been deleted, particularly those posted to publicly-owned television stations, that is, RTVE and France2.

Regarding the second specific objective, concerning dialogic principles, the inclusion of useful information was more intense in the cases of Telediario, Canal 24 h, Antena3 Noticias and la Sexta Noticias (Spain), TF1 Info and France Info (France). The generalist profiles (RTVE, Atresmedia, TF1 and France2), used self-glorification strategy, with self-promotion elements about the debate, its broadcasting times, its staging or its sections, good examples of the non-existence of dialogic loop preservation. As fot the generation of return visits, all the profiles used hashtags (#DebateRTVE, #ElDebateDefinitivo, #DebateAtresmedia, and #EleccionesGenerales in Spain, and #DebatMacronLePen, #LeDebat2022 and #Présidentielles2022 in France), frequently, and combined them in their posts, managing to bring many users back to their profiles. All the profiles analyzed met the dialogic princi-

ples concerning useful information and return visit generation, via their mentions of the candidates and parties, as well as the above-mentioned labels. Nevertheless, no interest was shown in generating conversations between televisions and users.

As for the third specific objective, to examine whether the agenda-setting of the television stations matched the main interests of the Twitter users, this research revealed that the content posted by the journalists, and the headlines highlighted by the television stations did not match, in most cases, the criteria of the users. This gave rise to divided opinions within the audiences, who often questioned the selection criteria of the media representatives. Memes and mocking tones were frequent in the user responses.

This information confirms the hypothesis that states that the dialogue and the conversation about the content posted by the different television stations, analyzed on the occasion of the televised election debates, does not exist, as confirmed by previous research papers on other candidate debates in the past.

Further research could seek to explain how politicians can engage in conversations with their voters while participating in election debates, or to determine cultural differences and similarities among different countries, considering both border proximity, and cultural and linguistic proximity. Nowadays, social media deliver messages to potential voters in a quick and effective way, and politicians are very aware of that fact; this could lead to more open dialogues and possibly to an online election debate system where the citizens participate through the classical direct question model.

**Author Contributions:** Methodology, J.F.-P.; Investigation, C.M.-B. and T.R.-M.; Resources, T.R.-M.; Data curation, J.F.-P.; Writing—original draft, J.F.-P.; Writing—review & editing, C.M.-B.; Supervision, J.F.-P. All authors have read and agreed to the published version of the manuscript.

**Funding:** This article is a result of the projects: "Fakelocal: Map of disinformation in the Autonomous Communities and local entitie of Spain and its digital ecosystem" (Ref. PID2021-124293OB-I00), funding by the *Ministry of Science and Innovation*, the *State Research Agency* (*AEI*) of the *Government of Spain*, and by the *European Regional Development Fund* (*ERDF*) of the *European Commission*. "DEBATrue: Fight against misinformation and value criteria in electoral debates on Television and Digital Media: verification platform and blockchain" (PDC2021-121720-100), of the Spanish *State R+D+i Program to carry out concept tests*, within the *State Program Challenges of Society* and the *State Program for Scientific, Technical and Innovation Research*. "Valcomm: Public audiovisual media before the platform ecosystem: management models and evaluation of the reference public value for Spain" (PID2021-122386OB-I00), funded by the *Ministry of Science and Innovation*, the *State Research Agency* (*AEI*) of the *Government of Spain*, and by the *European Regional Development Fund* (*ERDF*) of the *European Commission*.

**Institutional Review Board Statement:** Not applicable.

**Informed Consent Statement:** Not applicable.

**Data Availability Statement:** Compiled data has been included in methods and results sections.

**Conflicts of Interest:** The authors declare no conflict of interest.

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
