# Peer review of "Use of Twitter during Televised Election Debates: Spanish General Election (28 April 2019) vs. French General Election (24 April 2022)"

_societies, doi:10.3390/soc13030070_

Round 1

Reviewer 1 Report

The topic of the article is interesting for the field of knowledge. Despite focusing on a very specific aspect, such as the use of Twitter during televised election debates, the results fill an existing gap in the previous literature. In addition, the focus on two countries allows us to arrive at interesting results, by way of comparison, which could also be applied to other countries and future electoral periods.

The general presentation of the article is adequate. On the one hand, the abstract presented is correct and clearly identifies what is the topic, objectives, methodology and main findings of the research. The body of the text follows the same structure, being able to clearly identify all these points. Thus, we observe that the literature review correctly contextualizes the topic of study. Likewise, it uses relevant and very recent research that allows us to offer a broad x-ray of the subject. The objectives are clearly identified in the body of the article and relate to a clear and interesting hypothesis. The methodology is well explained and detailed and transparent because it clearly indicates and defines the analysis model. The sample is correct, and its choice has been sufficiently justified. The number of channels and programs analysed seems sufficient and valid to respond to the objectives of the research. The weakest point of the article, however, is found in the results. Despite being clearly stated, it is very descriptive. The authors only limit themselves to describing the different data extracted, which are already observed in the tables, and to highlight some related ideas. Especially in point 3.1. On the contrary, the authors fail to raise or identify future challenges or trends based on the results obtained. Therefore, the conclusions tend to be a summary of the results, so the article does not make substantive and sufficiently novel contributions.

Although it is a exploratory study, it is considered that the results presented are interesting and describe a phenomenon that can serve as a basis for future work focused on knowing the use of social networks such as Twitter in politically relevant events, as in this case, electoral debates.

Reviewer 2 Report

The article is interesting in its approach and helps to discover new communication trends and new ways of connecting with society in the field of political communication.

In the Spanish political landscape, in the first paragraph, it is defined as a two-party system. Third parties in discord have long played a very important role. Currently it should be defined as an imperfect two-party system and that has many political connotations (there is abundant bibliography on the matter).

Section 4 should be called Discussion of results and conclusions

In the discussion of specific objective 2, the differences are marked in terms of the use of twitter by Spanish and French television. It would be interesting to include here some discussion about the different cultures of use of the platform in the different European countries (there is a bibliography on the matter).

On the other hand, an analysis of the political contexts is made, but the bibliography on the typology of debates by country and the repercussion that each type of debate has on the audience and therefore on the electoral results is not included either.

Although the subject is of interest and topical, the results are not significant and their discussion is poor.

Round 2

Reviewer 2 Report

The revision of the article is OK